# Impact of the COVID-19 pandemic and policy response on access to and utilization of reproductive, maternal, child and adolescent health services in Kenya, Uganda and Zambia

Shiphrah Kuria-Ndiritu[1]*, Sarah Karanja[2], Brenda Mubita[3], Tonny Kapsandui[4], John Kutna[5], Dona Anyona[1], Joyce Murerwa[1], Laura Ferguson[6]

1 Amref Health Africa, Headquarters, Nairobi, Kenya, 2 Kenya Medical Research Institute, Nairobi, Kenya, 3 Amref Health Africa in Zambia, Ndola, Zambia, 4 Amref Health Africa in Uganda, Kampala, Uganda, 5 Amref Health Africa in Kenya, Nairobi, Kenya, 6 Institute on Inequalities in Global Health, University of Southern California, Los Angeles, California, United States of America

* Shiphrah.Kuria@amref.org, shiphonk@yahoo.com

## Abstract

Global health crises can negatively impact access to and utilisation of essential health services. Access to and utilisation of reproductive health services were already challenged in Sub-Saharan Africa with the COVID-19 pandemic further complicating the critical situation. This cross-sectional qualitative study aimed to assess the impact of the COVID-19 pandemic and policy responses to it on the access to, and utilization of reproductive, maternal, child and adolescent health services in Kenya, Uganda, and Zambia. It sought to explore the perspectives of women of reproductive age (18–49), frontline health workers and government representatives, all from geographies that are under-researched in this context. Using purposive sampling, key informant and in-depth interviews were carried out with 63 participants across the three countries between November 2020 and February 2021. The study population included women of reproductive age (18–49 years), front-line health service providers, and government representatives We established that COVID-19 and the policy response to it affected access to and utilization of services in the three countries, the most affected being antenatal care, delivery, family planning, and immunization services. Women reported not accessing the health facilities for various reasons. Barriers to access and utilization of services cut across all the socioecological levels. Movement restrictions, particularly in Uganda where they were most severe, and fear of contracting COVID-19 at health facilities were the most reported barriers. Weak structures at community level and inadequate supply of commodities in health facilities exacerbated the situation. Mitigation factors were put in place at different levels. There is need to strengthen the health system, particularly the supply chain and to have services closer to the community to enhance access to and utilisation of services at all times and particularly during crises such as the Covid-19 pandemic.

**Data Availability Statement:** All relevant data are within the manuscript and its Supporting Information files.

**Funding:** The authors received no specific funding for this work.

**Competing interests:** The authors have declared that no competing interests exist.

## Introduction

The Coronavirus disease 2019 (COVID-19) pandemic affected all sectors directly or indirectly and has had far reaching effects on already overburdened health systems in many countries with significant implications on health worldwide [1]. History has shown that pandemics limit access to healthcare, with preventive and reproductive healthcare being affected severely [2, 3]. Health services such as reproductive, maternal, child, and adolescent health (RMCAH) services, treatment for hypertension, diabetes and their complications faced numerous challenges during the COVID-19 crisis, being partially or completely disrupted in many countries [4, 5]. These challenges have had negative impacts on people's health and quality of life, and put extra strains in the achievement of many global health indicators such as the sustainable development goals (SDGs) [6].

The pandemic occurred at a time when many Sub-Saharan African countries including Kenya, Uganda and Zambia were struggling to make progress towards the SDGs [6]. Maternal and child mortality and morbidity in these countries have remained high, due in part to weak health systems [7–10]. Many women do not have access to modern contraceptives; many adolescents and young people have unintended pregnancies, children, particularly the under-fives, continue to suffer high morbidity, most of which is preventable [8–10].

Numerous studies and reports have examined the impact of the COVID-19 pandemic on access to and utilization of RMCAH services, including emergency obstetric and new-born care, finding widespread disruptions in Africa [5, 6, 11]. Many health facilities faced challenges including diversion of resources to the COVID-19 response compromising the provision of regular services, leading to reduced access to and utilization of essential care [5, 11, 12]. Several studies reported a decline in antenatal and postnatal care utilization during the pandemic with fear of infection, transportation difficulties, and restrictions on movement being among the major contributors to this decline [11, 13, 14]. Access to family planning services and contraceptives was significantly affected, including with stock outs of contraceptives, reduced availability of services, and limited mobility, which resulted in increased unmet need for contraception and a rise in unintended pregnancies [15, 16]. Routine immunization programs in many African countries experienced service disruptions, limited vaccine availability, and reduced healthcare-seeking behaviour, which led to a decline in immunization coverage, potentially increasing the risk of vaccine-preventable diseases [6, 11]. Adolescents across Africa experienced school closures, limited access to sexual and reproductive health services, disruptions in mental health support, and increased vulnerability to gender-based violence [17, 18]. All of this has contributed to adverse health outcomes, including an increase in maternal and child mortality rates [6, 11].

Beyond the pressure exerted on already strained health systems in Sub-Saharan Africa by COVID-19, there was also substantial social and economic disruption caused by the pandemic [5, 19, 20]. Similar to the experiences of previous epidemics such as HIV, SARS and Ebola, COVID-19 has exacerbated a range of pre-existing inequalities including those relating to socioeconomic status and gender [2, 19, 21]. The stringent mitigating interventions against the pandemic, such as lockdowns which increased barriers to access to healthcare, placed a disproportionate burden on vulnerable individuals, exposing them to poorer health and other outcomes [22, 23]. Many people reported loss of businesses and livelihoods worsening their economic situation, increasing income-related inequalities [20, 24].

Studies have shown mixed results, on the impact of the pandemic on the utilisation of RMCAH services, particularly in sub-Saharan Africa with some studies showing reduction while others did not show significant change [5, 25]. The pandemic disproportionately affected women and girls in many countries [2, 19].

Kenya, Uganda and Zambia experienced many of the same challenges to access and utilization of RMNCAH services as described above. On the supply side, there was disruption of the supply chain [26]. On the demand side, movement restrictions, myths and misconceptions about COVID-19 were widespread among communities, and fear of contracting the disease stopped many people from accessing health services [5, 11, 14, 27]. Lockdown measures, restricted movement, and transportation challenges limited women's ability to reach healthcare facilities [11, 14]. Additionally, some of the measures instituted at the facility level to curb the spread of the virus affected the provision and uptake of services [17]. Vulnerable populations, including rural communities and low-income households in Kenya, Uganda and Zambia were disproportionately affected by the COVID-19 pandemic as well as the associated policy measures instituted to control the spread of the virus [6, 28]. A variety of measures was put in place at various levels to improve access to essential health services [11, 14].

Efforts have been made to document the effects of the pandemic and the subsequent response on RMCAH and the extent and the quality of implementation of continuity of care guidelines and policies but there are still communities whose stories have not been heard and additional documentation is important. The objective of this study was to explore the perspectives of services users and providers as well as health policy-makers on the impact of the COVID-19 pandemic and the associated policy responses relevant to RMCAH services.

Previous studies carried out in Kenya, Uganda and Zambia focussed on different geographical areas from this study, and we did not find studies comparing the three countries. This study also explored the perspectives and experiences of women and healthcare workers (HCWs) on health seeking behaviour, access and utilization of RMCAH services during the COVID-19 pandemic, which provides valuable information for efforts to make health systems more resilient and responsive during the recovery phase and for future pandemics.

## Methods

### Study design and setting

This was a cross-sectional qualitative study, comprising of interviews with different stakeholders. In Kenya, it was conducted in one referral hospital, two sub-county hospitals, three health centres (HC), and three dispensaries in Homa Bay County, in the Western part of the country. In Uganda, the study was conducted in the regional hospital, six district hospitals and selected three Health Centres in Lango sub-region, Northern region. In Zambia, it was conducted in one regional and referral Hospital, one Mission hospital, one rural hospital and 16 primary health Care (PHC) facilities in Ndola District of the Copper Belt Province.

The study population included women of reproductive age (18 to 49 years old) living within the selected study areas since January 2020 who were either pregnant at the time of the study or had delivered at home or at the health facility during the COVID-19 pandemic; front-line RMCAH services providers from health facilities within the study areas and government representatives from the three countries.

The study regions were purposively selected, and the eligibility criteria is shown in Table 1.

Participants were purposively selected based on their suitability to provide the desired information. They included nine frontline health workers, nine women of reproductive age who had needed services (delivery, ANC or child health services) for their children and three government officials involved in policy implementation, per country. The research team worked closely with the community health volunteers to identify women of reproductive age. Government officials were from both the national level and the regional level. Health workers included doctors, nurses/midwives and clinical officers. The eligibility criteria allowed us to collect data from participants with varied backgrounds and experiences.

**Table 1. Eligibility criteria for selecting the study areas and facilities.**

|  | Kenya | Zambia | Uganda |
|---|---|---|---|
| Study area | Homa Bay County | Ndola district of the Copper belt province | Lango region in Northern Uganda |
| Selection criteria | • Amref Health Africa has an ongoing RMNCAH project in the region<br>• High maternal mortality<br>• High prevalence of COVID-19 | | |
| Sampling of health facilities | The selected facilities were stratified to include facilities located in urban, rural and peri-urban areas. The facilities were further stratified to tertiary, secondary and primary health care facilities | | |
| Sampling of health facilities | Stratified sampling technique was used to select 3 sub counties within Homabay County to represent Urban (Homabay Town Sub County), Peri-Urban (Rangwe Sub County) and Rural (Mbita Sub County).<br>The health facilities were further stratified to Sub County Hospital, Health Centre and Dispensary in each of the selected sub-counties. | Data was collected from 6 primary health care facilities and one Rural level and 2 mission hospitals. All the facilities offered similar maternal, neonatal and Child health services including labour and delivery services. | . Thus Regional referral hospital (Lira), 2 general district hospitals, 2 health center (HC) IVs, 2 HC IIIs and 2 HC IIs. |

## Theoretical framework

The study was based on the social ecological model (SEM) which posits that human behaviour is influenced by the interaction with the environment and has five levels (intrapersonal/individual, interpersonal/community, and structural (physical and cultural] and policy) [29, 30]. We explored how factors in all the different SEM levels influenced access and utilisation of RMCAH services acting at the different levels of the SEM.

## Data collection methods

In-depth interviews were conducted with women of reproductive age to explore women's experiences and perspectives on their health seeking behaviour, access to and utilization of RMCAH services during the COVID-19 pandemic. Community health volunteers assisted the research team to mobilize the interviewees. The key informant interviews with front-line RMCAH service providers were conducted to assess their perspectives of the access to and utilization of RMCAH services in the context of the COVID-19 pandemic. Key informant interviews were conducted with policy makers at the national and regional level. Those involved in either the development or the implementation of the policies were targeted; in each country the person in charge of RMCAH at either national or regional level was interviewed. The interviews sought to understand the government officials' views on how policies and guidelines relating to COVID-19 affected the access to and utilisation of RMCAH services.

In Kenya and Zambia, the interviews were carried out face to face following the appropriate Ministry of Health's COVID-19 prevention guidelines. However, in Uganda, telephone conversations were carried out, because physical interactions were not allowed by the ministry of health. The interviews took about 30 to 45 minutes.

There were six research assistants (RAs) in each country (total 18 RAs), all of whom had at least a bachelor's degree in social sciences or public health, previous experience conducting qualitative interviews, previous experience working with or interviewing health care providers and knowledge of RMCAH services, and were residents of the county/region/district where the study was conducted. RAs were taken through three days of training and one day of piloting the interview guides. In Kenya, interviews were conducted in English, Kiswahili and Dholuo. In Uganda, all interviews were conducted in English, while in Zambia interviews were conducted in English and Bemba. All interviews conducted in local languages were transcribed and translated into English. The interview guides are attached as S1 Appendix.

### Data management and analysis

Qualitative data were transcribed and analysed thematically and iteratively. Content and thematic analysis of transcripts was done with the codes being generated both inductively and deductively in each country. Coding and analysis were done manually across the three countries. The defined codes were then organized and sorted by relevant themes for reporting. Subthemes were allowed to emerge from the data through an iterative process and codes were refined as needed during the analysis. Written notes captured by the researchers during the key informant interviews were reviewed to complement the transcripts. Emerging trends were analysed according to the research objectives using a critical-interpretive approach. Transcripts are available as S2 Appendix.

Data triangulation and verification was done by comparing responses from different respondents to identify similarity of themes and areas of (dis)agreement on issues.

### Ethical considerations

Ethical approvals were received from an Ethics and Scientific Review Committee (ESRC) in each country. In Uganda approval was given by the Makerere University College of health sciences school of public health higher degrees' research and ethics committee; Protocol 891 while in Kenya approval was by the Amref Ethics and Scientific Review Committee, approval number P853-2020 and in Zambia by the Tropical Disease Research Centre ethics Committee; FWA number 00003729. Approval from the National Commission of Science and Technology (NACOSTI) was obtained in Kenya, License No: BAHAMAS ABS/P/20/6877 and from the National Health Research Authority in Zambia; Ref No: NHRA00001/25/09/2020. The implementation of this research was in full compliance with human subjects' ethical requirements, and informed consent was taken appropriately. We excluded girls aged 15 to 17 years due to the anticipated challenge in seeking consent for their participation from parents/guardians especially during the pandemic period.

## Results

The results are organised according to the main themes. The first theme is the effect of the pandemic and the associated policies on the access to and utilisation of RMCAH services. The second, more extensive theme explores the reasons for decreased access to and utilization of services by the levels of the SEM; closely related to this one another theme that emerged was the quality of care and the experiences of the women as they sought services. Finally, the innovations to address the arising challenges are explored.

### Effects of the COVID-19 pandemic and the resulting policy responses on access to and utilization of RMCAH services

Across the three countries, HCWs observed a reduction in the number of clients attending ANC clinics, family planning, delivery services, immunization and child welfare clinics, particularly during the initial periods of the COVID-19 pandemic. Children defaulted on vaccinations and the child welfare clinics as pregnant mothers missed their ANC visits. The reduction in utilisation of these services was attributed to the effects of COVID 19 such as fear of contracting the infection and to the policies towards mitigation of the pandemic. These policies included movement restriction, transport regulation and restrictions on service provision.

> "*It (Covid -19) affected us by even the numbers of the people in that all services* (RMCAH) *offered lowered during the lockdown. . . . . . .. Delivery services also dropped because of the same issues*". (KII HCW, Uganda)

"*The number (accessing and utilising family planning services) has gone down. . .At first it was around 40s and recently it is 28 [per month]*". (KII HCW, Kenya)

"*We had a reduction in family planning coverage from 45% to 35.8%. That is contraceptive prevalence rate under reproductive health. So, those are the effects on family planning services.* (KII Health Official at the district, Uganda)

"*The numbers (of ANC clients) are not the way they used to be like way back. They have decreased*". Service provider (KII HCW, Zambia)

"*It (baby welfare clinic clients) has decreased because they opt to stay at home waiting for the immunization. . .. just a few come (for Youth friendly services) about three out of 50*. Service provider (KII HCW, Zambia)

In all countries, the pandemic led to late ANC bookings with some pregnant women starting ANC in their second and third trimesters. In Kenya, HCW0 reported that some women attended the first ANC visit and only returned for delivery. In Uganda, HCWs reported a decline in the number of new ANC clients with few women attending four or more ANC clinics. Clients in Kenya and Zambia reported booking the clinic late and reducing the number of visits.

"*COVID has really affected the turn up of clients. In the first days for example the expectant mothers would come for the first visit, from there you would see her during delivery.*" (KII, HCW, Kenya)

"*I just thought of going to book for antenatal care at 4 months*" (IDI, Pregnant Woman, Zambia)

## Barriers in accessing and utilising healthcare services during the COVID-19 pandemic by socio-ecological levels

As explored below, the reduction in uptake of RMCAH services was caused by barriers that cut across all the socio-ecological levels. At the individual level fear of contracting COVID-19, fear of being tested for COVID-19 and getting quarantined and general fears around some of the government measures were reported. At the interpersonal and community level, rumours and misconceptions were identified as barriers to access and utilisation of services. At the policy and systems level, movement restrictions that resulted in transport challenges, costs associated with access, and lack of supplies, commodities and services were the main barriers.

### Individual level barriers

**Fear of contracting COVID-19.** Fear of contracting COVID-19 was reported across the three countries as one of the major barriers in utilization of health services especially at the onset of the pandemic. Health facilities were viewed as a risky place where COVID-19 could be contracted given they are where every sick person including those with COVID-19 go to seek treatment.

". . .. .*people are fearing to come to the clinic, they fear it's the place where there is Covid and won't come to access the service*". (KII~ HCW, Zambia)

"*I started (ANC) at 5 months; I am now at 7 months. I initially had fears that I would contract COVID*" (IDI WRA, Kenya)

"*We are scared of contracting Covid 19 from the health facilities. I am only at the health facility because I was in extreme pain, and I had no way out*". *(IDI WRA, Zambia)*

Some women who delivered at health facilities reported being worried that they and their babies might contract COVID.

**Fear of being tested for COVID-19 and getting quarantined and general fears around some of the government measures.** Fear of being tested for COVID-19 and consequently quarantined was also reported in Kenya as another barrier in the access to health services in the facilities, particularly during the onset of the COVID-19 pandemic. In the initial phase of the pandemic clients were screened for COVID-19 and if it was suspected they might be infected, they were taken into isolation/quarantine.

"*Something that makes them fear more especially when the temperatures are high that they will be taken to a quarantine centre. So people just prefer buying drugs from a pharmacy to going to the hospital.*" (IDI~ WRA, Kenya)

In Uganda, negative perceptions of some of the COVID-19 prevention measures, particularly the temperature screening using thermo-guns was reported to be a hindrance in access to health care services for some of the community members.

"*. . . a man who was bringing his children for immunization . . . .. he said no I do not want to be gunned, that is an evil practice, so he went back.*" (KII~ HCW, Uganda)

## Associated cost of accessing and utilising the services

While most of the health services are free at the health facilities, it was reported that some clients are increasingly financially constrained due to COVID-19 and unable to afford the small administration fees charged, or the record books required.

"*It's getting harder and you have to buy a book while there is no money. For some people even twenty shillings (less than 0.2 USD) to buy a book becomes difficult to get because of Corona. So they don't go to the hospital.*" (IDI~ WRA, Kenya)

In Zambia, some of the respondents indicated that they incurred costs, such as that of photocopying the ANC card, which they felt were expensive given the tough economic times. The general effects of the pandemic such as economic hardships made it challenging to access and utilise health services

"*Life is hard, business is so slow in this COVID era, and my husband lost his job because of COVID-19. I wanted to deliver from the facility, but circumstances made me deliver from home. . . . .. I had no money for a taxi. I delivered on my own in my house assisted by a volunteer from the clinic.*" (IDI_ Zambia)

## Interpersonal and community level barriers

**Misconceptions about COVID-19 at community level.** There were misconceptions that health care workers had contracted Covid and could infect those coming to seek health care services.

"*The barrier was the thinking that all health workers had COVID and all those who were coming to the facility were sick with COVID.*" (KII_HCW_, Uganda)

Even when outreach services resumed, there was still reluctance of the community members to use the services for fear of contracting COVID-19.

"*we used to go for an assessment of nutrition in the communities but the challenge we are getting now, if we move to the community, they will run away because of the fear of the pandemic. This is the same with immunization, it has reduced our coverage on immunization because people think we have taken them COVID-19 and run away.*" (KII~ HCW, Uganda)

### Policy and systems level barriers

**Movement restrictions and transport challenges.**    A major barrier in the access to health services was the movement restrictions, which were introduced in all the tree countries. In Kenya, the government introduced a night curfew at 7pm in the initial phase of the COVID-19 pandemic, making it difficult to access health facilities at night as there were no transport services. Additionally, people feared arrests and other consequences of encounters with the law enforcement during the curfew hours. Movement restrictions were particularly severe in Uganda: the government introduced a lockdown and a curfew accompanied by transport restrictions in the early phase of the pandemic in March 2020. Lockdown was eased but reintroduced in June 2021 when there was another wave of infections. The result was difficulties in access to health services not deemed emergencies including some RMCAH services.

Some women had planned to deliver at the health facility but ended up delivering in the community due to the COVID-19 associated restrictions such as movement restrictions and night curfews. There was unclear information on the policies and their implementation, with some women reporting that their delivery at home was due to inadequate information on how to access health facilities during curfew hours. The women did not get clear information that in case of a heath emergency, they could be allowed to go to the facility despite the curfew. The women were uncertain if they would be victimised as they tried to get to the facility for emergencies. These circumstances led to women delivering at home despite having planned to deliver at the health facility.

"*. . . The challenge comes especially with the April 20th restriction of movement. . . . .., labour does not follow the restrictions in those hours. It will come at midnight but then you are not supposed to move so what do you do*? *Do you deliver at home*? *Do you risk moving to the facility*? *Are you going to be arrested by the police*? *So those people who don't know their rights because if you are in labour you can pass the police roadblocks but there are others who don't know so they would rather stay at home. . ..*" (KII_, Kenya)

"*. . .if you look at most of our emergencies from all the data that we've. . .., most emergencies happen in the evening or at night. . .. The curfew is at 7 p.m.; all taxis have gone home, everybody that you know with a motorbike or the boda boda fellows who could help you like before are all at home because of the fear of unknown.*" (*KII Kenya*)

"*I delivered at home. . .it was late. . . around 8pm in the evening and you know the issue of transport even was a problem. By that time, you could not find a boda-boda (motorbike transport service) even if you searched for one. . . I failed to get transport, I sent for a traditional birth attendant who lives within [the community].*" (IDI WRA, Uganda)

"*We accepted the curfews imposed by the government and we were kept indoors, I personally felt intimidated by these…..*" Adolescent WRA, Zambia).

"*If we could have been told that there would be no problem on the way at night we would have just reached the hospital and delivered there. We would not have delivered at home or along the way.*" (IDI_WRA, Kenya)

In Uganda, the challenges in accessing transport, especially during the lockdown, resulted in birth complications, with some cases of maternal and neonatal mortalities being reported. Even in the cases of emergencies, the bureaucracy in securing transport permits from Resident District Commissioners (RDCs) or District Health Officers (DHOs) was a challenge. As one of the HCW noted:

"*. . . You need to get a letter from either the RDC or the DHO in order to access the road to go where you were supposed to get the service from.*" (KII~ HCW, Uganda).

In Uganda health workers reported that during the lockdown women on short-term family planning methods such as pills and injections lacked transportation means to access health facilities leading to interruptions in family planning use and unintended pregnancies among young women.

"*When the lockdown started there was no transport from home to wherever they could get these services [family planning services] and majority never went back for refills or injections. As a result, women and young girls conceived and we lost some.*" (KII_HCW Uganda)

"*Family planning was reduced, as you know family planning is not an emergency to some people, and then they would say aaah-aaaah. [Meaning no] all those services were reduced.*" (KII, HCW, Uganda).

In Zambia, movement restrictions mainly affected the availability and cost of public transport. Cost of transport was reported as a barrier to accessing and utilising services across all three countries due to the limited transport availability and the guidelines imposed on the sector. This situation was particularly difficult for communities because of the tough economic environment resulting from the pandemic and the accompanying restrictions.

"*There is low access and utilization due to transports costs, sometimes these are the reasons why our women are delivering from home. They are having a hard time sourcing for money because of the pandemic*". Service provider Zambia)

"*. . . The issue that the vehicles have to carry the required number of persons; the Nissans carry 14, and now they are supposed to carry eight (8). There is that tendency of the vehicles to increase the rates. The motorbikes were supposed to carry one person.*" (KII~ County health official, Kenya)

With transport costs high, some clients choose to access lower level health facilities or pharmacies rather than travelling further to access larger health facilities with more capacity.

"*Transport costs are very high for them, remember that the government came with a policy that you have to space, somebody in a taxi has to pay double, boda-boda (motor bikes) are now very expensive to bring someone to a regional referral here. They will resort to small units or peripheral units, other clinics or drug shops calling them. So, by the time they are referred*

*here they are in bad state because they delayed as they were thinking of reducing cost by remaining home."* (KII, HCW, Uganda)

*"The biggest challenge now is that at the regional referral, we are receiving these mothers late, and this is our biggest challenge. Sometimes they are in their dying moment and they have very bad complications. Sometimes the babies have already died, and that is the biggest challenge. . ."* (KII~ Government Official, Uganda)

## Shortage of supplies, commodities and capacity to offer services

Limited supplies, and commodities including medicines, vaccines and personal protective equipment was cited as a barrier to accessing health services across the three countries, even if clients could get to health facilities.

*". . . there were short supplies of medical supplies because they could not enter the country. In some cases they [clients] would withdraw and say why am I going there when I am not getting the services I need."* (KII~ HCW, Zambia)

*"Sometimes people go to the health facility but they fail to get drugs, so going there again becomes difficult for them. Like there is a woman who told me that she took the child to (name) Health Centre but she did not get the BCG so she had to go to (name) facility. It discourages clients."* (IDI_WRA, Kenya)

*"The biggest challenge here is the issue of supplies, there are key things that must be there if you are to maintain some quality and those things are given to us by NMS (National Medical Stores) or MOH and if they do not give, then there you're compromised and there is nothing much you can do. . ." (*KII, HCW, Uganda)

It was also reported by some key informants that the health care workers were not well prepared to offer services in the context of the pandemic. Although there were trainings to equip some HCWs, coverage was low. In addition to limited information and knowledge, personal protective equipment shortage was a challenge.

*"We are talking about 75 health workers–one per facility–which means we only cover 75 facilities but we have 283 health facilities so we need more trainings."* (KII, Kenya)

*"There was fear. You know when there are no guidelines. . . . . . . There was fear and getting PPEs was a problem. . . . . . . . They don't have PPEs. We saw in Kakamega people [HCWs] ran away. I don't blame them because it is a disease which you don't understand, you sort yourself out first."* (KII_, Kenya)

*"We need a full package of PPE for MCH because these are very important services that we cannot avoid. They should be treated like people in the treatment centres so that we can work with confidence."* (KII_HCW_TR, Uganda)

## Interruptions to service provision

The HCWs reported that some services were actually stopped at some point due to directives from the respective authorities.

*"For OPD [Out Patient Department] services, what happened here at the beginning of COVID the unit was transformed into the COVID isolation centre and the OPD was*

*transferred to another place. This made people get lost while others feared coming because of COVID." (KII_HCW, Uganda)*

"At some point we were told to stop (outreaches). I think that was before June [2020]. We didn't have enough information regarding the pandemic . . . . . . . . ." (Service provider, Zambia)

". . ..*the community outreaches that we used to have, it was first stopped and up to now, we are not doing that integrated outreach, we only do for immunization. In addition, this has just started of recent just like about 2 months ago but it had stopped for some 4 months during the total lockdown." (HCW, Uganda).*

The HCWs were concerned about their safety and feared being infected with Covid. They made adjustments to reduce the number of clients coming to the facility for services thus compromising the access to and utilisation of RMNCAH services. They gave the clients return dates after longer periods. It was reported that in some instances, the HCWs actually declined to get close to the patients.

. . .. now during the pandemic you are like "if I make them to begin coming like the way they used to come, they would have overcrowded and if they over crowd I may get the infection. Therefore, we would give them two or 3 months for them to come back; the worst was one month but it used not to be like that". [HCW, Uganda]

"*Even in immunization, the ones who are getting the third doses, we tell them to come when their children are 6 months to minimize movements." (KII_HCW, Kenya)*

The provision of youth friendly RMCAH services, which are critical for the provision of contraception to young people, was interrupted during the initial phases of the COVID-19 pandemic. In Uganda, at the onset of the pandemic, the facilities were not offering the youth friendly services as they were considered, by the government, non-emergency.

"*It has changed, as I said earlier youth are no longer coming to attend this clinic for the clear reason that majority of them think that they are not sick. . .so they don't come to the clinic. What brings a person to a hospital (*during the pandemic*) should be an emergency you cannot avoid"*. (KII~ HCW, Uganda)

In Kenya, the youth friendly services also stopped even as the need for them increased as schools were closed and young people had more leisure time than usual.

"*Yes we have a room for the youths. . . With COVID-19 it stopped for a while. . . .. As we speak we have booked them for December. They used to have plays, songs and teachings. We reduced the activities due to COVID-19." (KII~ HCW, Kenya)*

"*I think the younger age adolescents suffered more because you even find that during this Covid period most of the school going adolescents conceived a lot and we have high numbers of students who are pregnant. . ." (KII HCW, Kenya)*

### Restrictions within health facilities

With regard to their experience attending ANCs during the COVID-19 pandemic, the WRA across the three countries reported that there were changes in service delivery in line with the COVID-19 guidelines and regulations. Those seeking services at the health facilities were

required to fully comply with the COVID-19 infection prevention measures such as hand washing/sanitization, wearing of face masks and social distancing. Failure to observe the COVID-19 guidelines, led to being denied access to services; clients were sent away from the health facilities. The cost of face masks was prohibitive to some. As such across the three countries, those who did not have face masks were restricted from accessing and utilising health services.

*"Yes. . . We were told to put on the masks. Those without masks were sent back home".* (KII~ WRA, Zambia)

*". . .When you go to the health facility [to deliver], you have to follow the routine (SOPs) for COVID, and if you did not follow the routine for COVID they do not work on you or else they delay to work on you."* (IDI WRA, Uganda).

In Kenya and Zambia, the requirement to have face masks led to sharing of masks or using unsuitable substitutes, thus defeating the purpose of enforcing wearing of masks for COVID prevention.

*"Some of them can't even manage a mere mask they would come and just put on a hanky it will keep on falling off from their face they pick it up. Sometimes the mothers themselves exchange masks because you put up a policy to say if you have not put on mask you will not be attended to. . .."* (KII~ HCW, Zambia)

In Kenya, women reported that it was difficult to adhere to the COVID-19 preventive measures such as maintaining social distance (due to space limitations within health facilities) and wearing a face mask during delivery (it was uncomfortable); only the nurses had their face masks on. They also reported minimal interactions with HCWs. People without masks were being arrested thus further discouraging those who could not afford them from going to the facilities.

## Quality of care

Women reported different levels of satisfaction with the care they had received at health facilities. Women in Kenya and Zambia reported having had pleasant experiences though some others in Zambia, but from different locality, reported unpleasant ones. In Uganda, some women reported delays in being attended to for those who did not adhere to the COVID-19 prevention measures while some in Zambia reported that the services were fast. It was reported that even heath workers were afraid of offering services.

*"The nurse treated me well even if it was a weekend. I could not wear a face mask but the nurse had one on. She treated me with respect and I appreciate."* (IDI WRA, Kenya).

*"The way the health workers mistreat people at the facility also scares away people"* (IDI~ WRA, Uganda)

*"With the service, we were sitting two per bench, they weighed the babies, and blood tests were done. The service was very fast."* (IDI Kawama HC, Zambia)

*". . . I have no card; I am afraid they will shout at me for not having a card."* (IDI_WRA, Zambia) Lubuto HC

". . .I went for immunization, and we stayed there for some period, and nurses were not there. Then afterward we heard rumours that they feared to attend to us because of COVID-19." (IDI, HCW, Uganda)

"I think there is need for more knowledge because lack of knowledge is what is causing fear. We encourage the members of staff to adhere to what they know about Covid 19. Through counselling and creating awareness of Covid 19, fear will constantly be dealt with and service delivery will continue normally". (Service provider Zambia)

When women were asked if they received sufficient information on COVID 19 and on the services they were receiving, they did not feel they received much information on the pandemic, including from HCWs.

"Not really, we were only encouraged to maintain social distance and to wear face mask as we were". (WRA, Zambia)

According to health workers, provision of information to clients was limited due to COVID; gathering in groups was discouraged which led to missing out on clients receiving information from the health workers and from fellow clients. Health care workers did not get sufficient training to equip them with the information they needed to share with the community members.

". . ... since we were not holding gatherings, health education of clients was not being done. Sometimes we did not respond to patients' questions because we never wanted to get close to people with COVID like symptoms." (HCW, Uganda.)

"We didn't have enough information regarding the pandemic and what to tell the community." HCW Zambia

## Alignment and structural innovations to address challenges

In a bid to make services accessible despite the challenges occasioned by the COVID pandemic, various measures were put in place, some using the existing structures particularly at the community level while in other cases new approaches were created. The facilities worked with existing community structures and partners to reach the community. In Kenya, the HCWs reported using Community Health Volunteers (CHVs) to help track some of the clients who had defaulted on immunisations. To overcome curfew restriction challenges, authorities came up with letters to issue to the clients, transporters and relatives as they left facilities late after receiving services. In Zambia, CHVs as well as other cadres were used to track the pandemic and to reach out to the community with information.

". . . In Homabay County we came up with an authority letter which we were giving every facility that when a motorbike drops a mother who is in labour at night then they are given to go home." (KII, Kenya)

"The management of COVID is within the structures that are already in existence; structures such as community health volunteers, health centres/health posts statistic health office, the PHO [public health office]. I know that at the PHO there is an incident management system that responds to the progress of the COVID disease." (KII, Zambia)

"We have the community based organizations that we were involving. We have NGOs that are working with the community. There are some women's groups that were mounting

*vehicles with the mega phones . . ...going from one place to another. . .., giving the messages. . . We also had our health promotion officer here and the county working . . .. in the county and sub-county."* (KII_ Kenya)

*"During the lockdown, there was an intervention done by RHITES-NORTH-LANGO (*an NGO*) in conjunction with MOH where we used to carry medicines to the community." (HCW Uganda).*

*"We have that bit of taking the drugs to the clients through the CHVs. They form a group . . .. .. . .. and choose one person who can collect for them the drugs or the CHVs."* (KII_ Kenya)

In Kenya, healthcare providers emphasized use of long-acting reversible contraception methods as a strategy to alleviate the burden on healthcare facilities. For individuals who preferred short-term contraceptive pills, a standard three-month dosage was typically provided. However, if individuals demonstrated stability and adequate understanding, the supply was extended to six months.

*"We are now stressing on long term family planning methods whereby we don't interact with the patients soon. Even the short term like the pills, we are able to give many. We usually give 3 monthly, but we can go up to 6 if they are stable and educate them".* (KII HCW, Kenya)

In Zambia, it was reported that the government had already implemented self-injectable contraceptives before the pandemic. This family planning method would have been an alternative that would have enabled women to autonomously manage their contraceptive needs without the necessity of in-person visits to healthcare settings and thus would have alleviated congestion in healthcare facilities. However, the commodities were out of stock and hence did not work when needed most during the pandemic.

*". . .The government also introduced a new self-injection for family planning so that mothers do not come to the clinic for this service. They inject themselves at home. . . However, that type of family planning is in short supply. We have not had it for eight months now. It's not there."* (KII, HCW, Zambia)

### Looking across all socio-ecological levels

It is clear that a wide range of factors across all the socio-ecological levels influenced access to and utilization of RMCAH services, with some arising from the pandemic itself and others a result of government responses. All of these factors interacted to create a constrained environment for both WRA and HCWs who had inadequate information and resources to respond appropriately. Table 2 below summarises the different barriers to access to and utilization of RMCAH services across the barriers the socio-ecological levels.

## Discussion

### Access to and utilisation RMCAH services

Our findings indicate that accessing and utilising maternal health services such as antenatal care, facility deliveries and family planning services during the COVID-19 pandemic in Kenya, Uganda and Zambia were compromised and reduced. The reduction occurred through multiple pathways; they have been explored according to the SEM levels. Both the service

**Table 2. Barriers in accessing and utilising healthcare services during the COVID-19 pandemic by socio-ecological levels.**

| SEM levels | Barriers in accessing and utilising healthcare services during the COVID-19 | Country reported |
|---|---|---|
| Individual | • Fear of contracting COVID-19 | Kenya, Uganda and Zambia |
| | • Fear of being tested for COVID-19 and consequently quarantined | Kenya |
| | • Fear of thermo-guns | Uganda |
| | • Associated cost of accessing and utilising the services | Kenya, Zambia, |
| Interpersonal and community | • Limited information and misconceptions about COVID-19 at community level | Zambia, Uganda |
| Policy and Systems | • Movement restrictions and transport challenges | Kenya, Uganda and Zambia |
| | • Shortage of supplies, commodities and capacity to offer services | Kenya, Uganda and Zambia |
| | • Interruptions to service provision (some services were not being offered) | Kenya, Uganda and Zambia |
| | • Restrictions within health facility (no mask, no service) | Kenya, Uganda and Zambia |

providers and the clients corroborated the difficulties in accessing and utilising services, many of which were similar in the three countries. The main challenges were related to the policies that were introduced to mitigate the spread of the pandemic in all the three countries. At the individual level, fear of contracting the infection both by the WRA and the HCWs was reported. At the organisation level, limited ability to comply to the prevention guidelines was reported by many HCWs and WRA. Most of our findings corroborate the findings of other studies that have now been published as shown in the discussion that follows, even as we cover new geographies.

**Antenatal and delivery services.** The reduced access to and utilisation of services found in our study is similar to findings by Kotlar et al., 2021 and Tadesse, 2020 [17, 31]. Kotler and colleagues did a scoping review of published evidence globally while Tadesse did his study in Ethiopia. They established decreased access to and utilization of antenatal care services due to COVID-19 pandemic and the resulting policies. A study done to assess the utilization of antenatal care and facility deliveries among refugees in Kenya indicated decline in access to and utilisation of these services attributed to the fear of contracting COVID-19 and economic challenges occasioned by the pandemic and associated policies [32]. A study in Uganda documented reduced access and utilisation to child health services [11].

We belief that the reduction in access to and utilisation of care posed a great risk to clients. Although our study did not collect data on maternal mortality, Roberton et al. [33] estimated an 8.3–38.6% rise in maternal mortality per month due to interrupted maternal and child health services during the COVID-19 pandemic in 118 low- and middle-income countries (LMIC). While we are unable to substantiate any such rise quantitatively from our study, it is likely that some deaths resulted from the inability to access care as reported by some respondents in our study. The World Health Organisation has indicated a large discrepancy between estimated global excess mortality and reported deaths due to COVID-19 in 2023 [6].

**Family planning services and youth friendly services.** The findings in our study of reduced access to and utilisation of youth friendly services through which the majority of youths and adolescents access family planning services imply the possibility of the vulnerable being disproportionately affected. These findings are, similar to other studies in Africa and

beyond, which have shown the vulnerable were disproportionately affected by both the pandemic and the mitigation policies, with many young women unable to access family planning services leading to increased unintended pregnancies as well as unsafe abortions [6, 15, 19]. Evidence indicates that there is high unmet need among married adolescents aged 15–19, and among women from rural areas, the poor, those with low education and those not exposed to mass media [34]. Though our study did not include adolescents aged below 18 years, the access challenges applied to clients of all ages accessing RMCAH in the area studied. By clients not accessing family planning services, then their need for modern contraceptive methods was not met.

Khowaja and Shalwani, reported that there was reduced access to and utilization of family planning services in health facilities resulting from COVID-19 pandemic and policies imposed to curb spread of the disease in Pakistan [16]. According to UNICEF, disruption in the provision of family planning services among young people, increased unintended pregnancies, childbirth complications, and child mortality [18]. Increased unmet need for modern family planning methods threatens the achievement of the SDG target 3.7. (ensure universal access to sexual and reproductive health-care services, including for family planning, information and education) which is tracked by the indicator 3.7.1, "Proportion of women of reproductive age (aged 15–49 years) who have their need for family planning satisfied with modern methods [35].

Similar to our findings on the interruption of provision of youth friendly services, Compact for Young People in Humanitarian Action and other studies found interruption of youth friendly services [11, 36].

**Postnatal care services including immunization and child welfare clinics.**   Disruption in immunization services has been reported in other studies. Ninety-five percent of countries in South-East Asia and Western Pacific reported disruption of routine immunization service provision due to COVID-19; vaccinations had dropped by 39% by June 2020 [37]. The reduction in access to and utilisation of child immunization services was caused by very similar challenges as those recorded in our study: fears of infection, movement restrictions and limited healthcare access. Evidence shows that 28 out of 62 countries suspended house-to-house immunization during the pandemic [13]. A study in Uganda documented reduced access and utilisation of child health services [11]. In all the three countries reduced access to and utilisation of child health services was reported in our study.

**Barriers to access and utilisation of RMCAH services by socio-ecological level.**   The pandemic reduced access due to fear of contracting COVID-19 while the measures put in place to curb spread of COVID-19 had an even greater impact with inadequate information exacerbating the situation.

## Personal/individual level

Our study findings that WRA avoided going to facilities for fear both of the infection and of contravening the measures instituted to curb the spread of the pandemic are similar to other studies. Oluoch-Aridi et al. [14], reported that amongst women in informal settlements in Kenya some avoided facilities due to fear of contracting COVID-19 while Musoke et al, [11] reported fear as a reason for clients keeping away MCH services in Uganda.

## Interpersonal and community level

The fears were made worse by the inadequate information that individuals had including rumours and misconceptions. Incorrect information including misconceptions and rumours have been reported in other studies. In Ghana myths and misconceptions, on the causes of the

disease and vulnerability to the disease were reported, including that the hot climate in Africa inhibited viral replication and transmission of the corona virus [38]. In Uganda, misconceptions that COVID-19 was a disease of the white and the elderly was reported [39]. In Zambia the misconceptions included that the disease had been introduced by foreigners with ulterior motives, that it was a political gimmick by the government, and that it was due to radiation from phones, which made people sceptical about measures such as testing for COVID-19 and wearing masks 27. It has been noted that misinformation posed a risk of vaccine hesitancy and continued spread of COVID-19 infection among pregnant women [36, 37]. Low levels of adherence to the preventive and control measures towards COVID-19 in the community have been reported due to limited information [7]. Disinformation has undermined trust, intensified fear and led to inappropriate behaviour in respect to the control of the pandemic [40, 41].

## Policy and systems level barriers

Our findings indicate that in all the three countries, there were disruptions in the medical supplies including personal protective equipment, and interruptions in offering services which negatively affected the provision of services have been corroborated by other studies. Reports indicate that the pandemic disrupted the supply chain particularly in the low- and middle-income countries [26]. Unlike the findings in our study where the HCWs did not have adequate information on Covid, a study in Lusaka city and Chirundu international border town in Zambia reported that HCWs had correct knowledge on the disease including the prevention measures [27]. HCWs needed information to enable them offer services in the context of the pandemic. Limited information amongst the HCWs in our study might have hampered the quality of information and services available to community members through the health system thereby further disincentivizing attendance at services and affecting the quality of care of those who received services. Unsatisfactory experience by those receiving services could negatively affect the subsequent visits to the facilities.

At the policy level, various some of the policies instituted to control the pandemic were a hindrance to access and utilisation of RMCAH services in all the three countries but access challenges due to movement restrictions appear to have been more pronounced in Uganda where the restrictions were stricter and in place for a longer time, compared to Kenya and Zambia. Many respondents from Uganda referred to lockdown and movement restrictions that severely interfered with access and utilisation of RMCAH services. These findings are corroborated by studies that have reported negative impact of the lockdown on access and utilisation of MCH services in Uganda [11]. While it is very important to control pandemics, the effect of limiting access to essential services such as health and the economic impact of restrictions on the population must be carefully considered.

The "No Mask, No Service" policy in health services was a barrier since the masks were costly. The stay at home policy, additionally contributed to clients avoiding health facilities, as has also been found in other studies [11, 14]. Similar to findings in our study, inadequate information on the pandemic and on the policies and guidelines introduced to mitigate the spread of the infection at different levels, resulted in reduced access to and utilisation of services [42].

## Quality of care and the right to health

Wangamati and Sundby [43] observed that the already strained health system in Kenya, struggled to deliver quality care due to the increased demand occasioned by the pandemic. In our study, the women had different experiences while seeking services some of which raise concerns about the quality of care clients received and indicate impediment of the realisation of their right to health in the three countries. The requirement of social distancing led to limited

interaction with HCWs and the regular health education was either shortened or not done in some cases. The limited information indicates a challenge in the quality of services offered and might have contributed to women not having enough information for birth planning as some reported being surprised when labour came earlier than the expected due date. Birth and emergency preparedness is a critical aspect of ANC to ensure women get to health facility for skilled care during delivery or in case of an emergency.

From our findings, the right to health for the clients who needed RMCAH services in the area was infringed in some instances. For example, failure to comply with the COVID-19 prevention guidelines (wearing of masks) led to denial of services for the poorest who could not afford to buy masks, thus exacerbating inequities. It is the role of the government to protect all citizens to the extent possible from infringement of the right to health, particularly the vulnerable. In this case, this could have been addressed by providing masks for the clients who did not have them instead of sending them away from the facilities. Women also reported having failed to go to facilities for delivery since they were not aware this would have been allowed despite the restriction movements. Providing accurate information on how to access essential services in the phase of this crisis that necessitated movement restrictions would have avoided these barriers to access. The existing community structures were pivotal in mitigating the negative effects of the pandemic on the access and utilisation of MCAH services. CHWs and other resource persons were used to reach clients at the community level and shows the importance of resilient community structures.

Some of the experiences of living through the COVID-19 pandemic may have led to better quality of care; washing of hands is known to be an important strategy in infection prevention. The availability of hand washing infrastructure and the focus on the practice could mean better quality of care in regard to infection prevention. Experience in Zambia of faster services correlates to findings in Kenya where a study done amongst women living in the informal settlements in the Embakasi area in Nairobi City, Kenya where most women perceived improvements in quality of care due to short-waiting times and hygiene measures [14].

Financial hardships occasioned by the pandemic contributed to difficulties in access to and utilisation of services in our study, a finding substantiated by other studies that found these hardships to have increased in the three countries and negatively impacted healthcare seeking [14, 44]. These findings corroborate those of other researchers, adding to evidence that will be critical to consider while planning the recovery phase to support investing in resilient systems in order to avoid the system weaknesses that contributed to negative policy related ill effects experienced during the pandemic. Many mitigation efforts were implemented through the existing community structures such as the CHVs and community-based organisation. These structures need strengthening for sustainable access to and utilisation of RMCAH services routinely and during crisis.

## Study limitations

The main limitations of this study include covering a limited geographical area in each country and hence the results may not be generalisable. In addition, we excluded adolescents aged 15 to 17 years old due to the anticipated challenge in seeking consent for their participation from parents/guardians especially during the pandemic period, and so we did not document their experiences. However, though we cannot make deductions on the experience of this group, the information about the access challenges applied to all the clients. We had hoped to triangulate our findings with routine health data but were unable to do so due to concerns around missingness across the study settings.

## Conclusion

The COVID-19 pandemic and the resulting policy responses negatively impacted access to and utilisation of most RMCAH services including ANC services, skilled deliveries, family planning, youth friendly services, and postnatal care services such as immunization and the child welfare clinics. There was a reduction in the number of clients seeking these RMCAH services.

Lessons learnt can usefully inform pandemic recovery and future pandemic preparedness; from the findings of our study we highlight some lessons and recommendations.

There is a clear need to strengthen the health system, including health services at the community level to enhance access to and utilisation RMCAH services at all times including during crises that could limit movement. Government and other stakeholders must prioritise this agenda in a sustainable way. Our study has demonstrated the need to plan for emergencies considering how the different levels of the SEM interact to affect the lives of community members. Additionally, the study demonstrates the similarity of the challenges faced by the health systems in the region. It is key to ensure that all communities have well trained and resourced community resource persons including community health workers/volunteers and other community leaders who would be able to get correct information promptly and be able to share with the community members in case of a crisis.

Contingencies are required to ensure continuous service provision even in the context of a pandemic, which includes ensuring an adequate supply of necessary equipment such as PPE; provision of clear information and clear guidelines including on how those needing services could access them. Implementation of guidelines during emergencies must be monitored closely to mitigate the emerging unintended negative effects particularly on the health of the community.

## Supporting information

**S1 Appendix. Interview guides.**
(PDF)

**S2 Appendix. Transcripts.**
(ZIP)

## Acknowledgments

All the CHVs who supported us in identifying appropriate study participants and all those who supported us in data collection and analysis, thus the consultants who coordinated data collection in the three countries;

Kenya; Dr John Oyare and Mr. John Dave Molla
Uganda; Mr. John Bosco Waswa
Zambia; Mr. Tato Nyirenda

## Author Contributions

**Conceptualization:** Shiphrah Kuria-Ndiritu, Sarah Karanja, Brenda Mubita, Tonny Kapsandui, John Kutna, Dona Anyona, Joyce Murerwa, Laura Ferguson.

**Data curation:** Shiphrah Kuria-Ndiritu.

**Investigation:** Shiphrah Kuria-Ndiritu.

**Methodology:** Shiphrah Kuria-Ndiritu, Sarah Karanja, Laura Ferguson.

**Project administration:** Shiphrah Kuria-Ndiritu, Brenda Mubita, Tonny Kapsandui, John Kutna.

**Resources:** Shiphrah Kuria-Ndiritu.

**Supervision:** Shiphrah Kuria-Ndiritu, Brenda Mubita.

**Writing – original draft:** Shiphrah Kuria-Ndiritu, Laura Ferguson.

**Writing – review & editing:** Shiphrah Kuria-Ndiritu, Sarah Karanja, Brenda Mubita, Tonny Kapsandui, John Kutna, Joyce Murerwa, Laura Ferguson.

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
