## [Decision Letter · Decision Letter 0]

1 Mar 2023

PGPH-D-22-00887

Impact of the COVID-19 pandemic and policy response on access to and utilization of reproductive, maternal, child and adolescent health services in Kenya, Uganda and Zambia.

Dear Dr. Kuria-Ndiritu,

Thank you for submitting your manuscript to PLOS Global Public Health. After careful consideration, we feel that it has merit but does not fully meet PLOS Global Public Health’s publication criteria as it currently stands. Therefore, we invite you to submit a revised version of the manuscript that addresses the points raised during the review process.

Please note that we have only been able to secure a single reviewer to assess your manuscript. We are issuing a decision on your manuscript at this point to prevent further delays in the evaluation of your manuscript. Please be aware that the editor who handles your revised manuscript might find it necessary to invite additional reviewers to assess this work once the revised manuscript is submitted. However, we will aim to proceed on the basis of this single review if possible.

The reviewer has raised concerns that the study methdology has not been described in sufficient detail to enable replication of the study. Furthermore, the reviewer has also encouraged additional discussion on how the current study further contributes to scientific understanding in light of related literature.

Could you please revise the manuscript to carefully address the concerns raised?

We look forward to receiving your revised manuscript.

Kind regards,

Lucinda Shen, MSc

Staff Editor

Journal Requirements:

1. Please indicate the full and correct funding information for your study and confirm the order in which funding contributions should appear. 

2. In the online submission form, you indicated that "This being qualitative research,the interviews transcriptions are available inpassword proteced files in the Amref data base and can be availed by the corresonding author upon request." All PLOS journals now require all data underlying the findings described in their manuscript to be freely available to other researchers, either 1. In a public repository, 2. Within the manuscript itself, or 3. Uploaded as supplementary information.

Additional Editor Comments (if provided):

Reviewers' comments:

Reviewer's Responses to Questions

**Comments to the Author**

1. Does this manuscript meet PLOS Global Public Health’s publication criteria? Is the manuscript technically sound, and do the data support the conclusions? The manuscript must describe methodologically and ethically rigorous research with conclusions that are appropriately drawn based on the data presented.

Reviewer #1: Partly

2. Has the statistical analysis been performed appropriately and rigorously?

Reviewer #1: No

3. Have the authors made all data underlying the findings in their manuscript fully available (please refer to the Data Availability Statement at the start of the manuscript PDF file)?

Reviewer #1: No

4. Is the manuscript presented in an intelligible fashion and written in standard English?

Reviewer #1: Yes

5. Review Comments to the Author

Reviewer #1: A. General comments

1. Relevant data and citations are not provided to support key statements. Study rationale is not well put. The transition from what is known to the statement of study aim is weak.

2. The methods are not described fully and clearly. Criteria, rationale of key methodological approaches, and other important details to support the methods are not provided and/or not properly presented.

3. Important data are not provided which weakens the results section. Data are not presented in a manner that could optimize integration of main results. Distribution of respondents does not seem to be balanced across study areas/countries.

4. Despite bringing valuable data, the discussion does not integrate the study findings. Major findings of the study are not well linked to prior work. There is a tendency to restate the results. Methodological limitations are not discussed rigorously. Policy recommendations are not directly informed by the study findings.

B. Specific comments

I. Abstract:

1. Overall, it is known that COVID-19 and the responses to it affected somehow access/utilization, so it might be necessary to focus on what aspects of access/care were most affected, which sub-population, and where (which country/community/etc)/why/how, and whether there was some similarity across the studied countries.

2. What aspects of the health system are weaker and more fragile based on the study findings?

3. What do the study findings imply in view of what is already known?

II. Introduction:

1. More recent/robust data might be available already on some aspects discussed here, which should be used and cited to improved the rationale/context of the study, including:

a) Impact of COVID-19 on health inequality in Africa: for example: https://doi.org/10.1186/s12939-020-01361-7;
https://doi.org/10.3389/fgwh.2021.686984;
https://doi.org/10.2147/RMHP.S324554;

b) National progress across time on SDG related indicators: https://www.who.int/data/gho/publications/world-health-statistics

2. Why do the authors consider "Maternal and child mortality and morbidity in these countries ... unacceptably high"? Based on what criteria?

3. The reference provided (Case, A., Paxson, C., 2011) discusses only HIV/AIDS, so the corresponding passage should be adjusted accordingly, and/or additional data/citations should be provided to include other another pandemic/epidemic:

a) Can other outbreaks, such as Ebola disease, be included, since it says in lines 64-66 "past pandemics ..." but the source cited only presents data on the impact of the AIDS pandemic (i.e., single pandemic)?

b) If not, why?

4. Are not there robust (peer-reviewed) sources available (instead of a single media source from one country)?

5. In lines 70-79:

a) Can data (actual effects, not potential effects) be provided to support the statements? If not, why?

b) Can lines 74-79 be combined and integrated to the paragraph 64-73? If not, why?

6. Can lines 80-90 be integrated and improved to improve the way in which the study rationale is put?

7. Can data/citation be provided to support this statement: "The pandemic occurred at a time when many Sub-Saharan African countries including Kenya, Uganda and Zambia were struggling to make progress towards the Sustainable Development Goals (SDGs). "? If not, why?

8. Can the authors explain/provide data/citations to support this statement: "Too many women do not have access to modern contraceptives; too many adolescents and young people have unintended pregnancies. Children, particularly the under-fives, continue to suffer high morbidity and mortality, most of which is preventable." Why do the authors consider these to be "too many"? based on which criteria?

III. Methods:

1. Which guidelines does the study conform to? Why?

2. In lines 93-113 the authors describe the areas and facilities where the study was conducted:

a) What were the eligibility criteria that were used to select the study areas and facilities in each country and across countries? Can these criteria be provided as a table? If not, why?

b) What is the rationale of the eligibility criteria?

c) What are the implications of these eligibility criteria on the validity of the study findings?

d) Why was the age of participants restricted to 18-49 years old? Why were girls (15-18 years old) not included given the WHO definition of women of reproductive age (e.g., see here: https://www.who.int/data/gho/indicator-metadata-registry/imr-details/women-of-reproductive-age-(15-49-years)-population-(thousands))?

e) Can a flow diagram of the process of selection of participants be provided (to provide a full and clear picture of the whole process)? If not, why?

3. Which questions where asked during these interviews? Can these questions be provided as a table (or at least as a supplementary material), indicating what questions were asked to which type of participant? If not, why?

4. In which languages where the interviews conducted in each country (particularly to women of reproductive age (WRA))? Was translation of the questions and answers necessary?

5. Who conducted the interviews to each group of participants?

6. Was any computer program used to save, process, and/or analyze the data? If yes, which program? If not, why?

IV. Results:

1. Was there some trend observed in the effect of the pandemic or related policies on the "Access to Antenatal Care (ANC) and Delivery Services" by age and type of pregnancy (primigravida vs multigravida), etc?

2. Is the study assessing access to or utilization of "Antenatal Care (ANC) and Delivery Services" or both? For example, see here: https://doi.org/10.1586/14737167.6.6.653.

3. Was reduced "Access to and Utilisation of Family Planning Services" reflected in terms of unplanned/unintended pregnancies?

a) Why?

b) was that assessed in the current study? Why?

4. What were the consequences of the reduced "Access to and Utilisation of Youth Friendly Services" epidemiologically?

5. Did the HCW specify for how long were these services interrupted? (i.e., what is the meaning of "for while"?) Was this uniform across all study areas/countries?

6. Were these "Immunization and Child Welfare Clinics" closed or only "clients" stopped using them or both?

7. Why "Movement restrictions were particularly severe in Uganda" compared to the other study countries?

8. Can "access" in line 199 be replaced with "use"? If not, why?

9. Was the "fear of contracting COVID-19" related to accessibility of anti-COVID-19 protection tools, such as masks, or other vulnerability factors? Was this assessed in the study? If not, shy?

10. Could at least one respondent from each country or study area be provided for each applicable aspect in the sub-theme?

11. Could a table be provided to summarize the data/main findings across sub-themes to facilitate integration of the results?

12. Why does Zambia appear to be relatively less represented and Kenya more represented? What is the implication of that?

13. "While most of the health services are free at the health facilities": where/which services/which facilities? Private sector/public health/all countries/all services?

14. Can national currency be exchanged to an international currency (such as international dollar) to give more clarity of the weight of the financial barriers given the costs of living in each country, where applicable? For example, see here: https://datahelpdesk.worldbank.org/knowledgebase/articles/114944-what-is-an-international-dollar.

15. Could some socio-demographic data of each respondent (at least of WRA) be given to facilitate understanding the context and whether there is some pattern across socio-demographic groups? If not, why?

16. Are these "fears" of "bad experience with the health workers" new (only in the context of the COVID-19 pandemic)? Did the COVID-19 pandemic and/or related response measures change these?

17. Why "there was unclear information on the policies and their implementation."? Which aspects were not clear?

V. Discussion:

1. Can "Access to RMCAH services" and "accessing services" be replaced with "Access to and utilization of RMCAH services" and "accessing and utilizing services", etc as indicated elsewhere? If not, why?

2. Can it be indicated that the pandemic reduced access and utilization through multiple pathways, and then indicate those that WRA and HCW identified as more significant/prevalent, and note whether these were similar across countries, and contrast that with previous work?

3. How do the findings of the current study relate to those of previous studies, such as: https://doi.org/10.1186/s12992-021-00666-8;
https://doi.org/10.1016/S2214-109X(21)00079-6?

4. Can it be discussed that the current study did not include women aged "15-18" years in the context of lines 385-392 and describe implications of that in the context of the study findings and the literature?

5. Was only "utilization of family planning services" that was "reduced"? Written this way, it might imply that the WRA had access but did not use. Again, for example, see and possibly discuss: https://doi.org/10.1586/14737167.6.6.653, and/or other applicable articles.

6. "unwanted pregnancies" or "unplanned pregnancies" or "unintended pregnancies"? Perhaps using the same terminology in the manuscript might be desirable to ensure consistency. For example, see here: https://www.who.int/news/item/25-10-2019-high-rates-of-unintended-pregnancies-linked-to-gaps-in-family-planning-services-new-who-study.

7. Can the SDG Indicator 3.7.1 on Contraceptive Use and SDG target 3.7 be referenced/discussed in lines 385-386, etc? For example, by indicating which aspect of SDG target 3.7 was most affected and/or indicated as being more affected/significantly affected in which country, etc, and relating that to world health statistics on SDG? For example, see: https://www.un.org/development/desa/pd/data/sdg-indicator-371-contraceptive-use;
https://www.un.org/en/development/desa/population/publications/pdf/family/familyPlanning_DataBooklet_2019.pdf;
https://www.who.int/data/gho/publications/world-health-statistics

8. "Restriction of movement hindered supply of family planning commodities affecting access to and utilization of FP hence increased unplanned pregnancies among the youths.": where? What is the significancy of this passage here?

9. What are the findings of other studies that studied the same topic in other African countries? How these align/relate to the findings of the current study?

10. Were the "Misconceptions and rumours for example about the thermo-gun" observed in all study countries and across all socio-demographic groups in the current study? How does that relate to what is already known and quality of information? What aspect of the quality of information was poor? Is there research on these aspects? Can these be discussed in light of the study findings?

11. "The right to health was infringed in some instances.":

a) Of whom?

b) How about "The right to health" for those whom would have been infected with SARS-CoV-2 and potentially die due to non-compliance by "clients" to basic COVID-19 prevention guidelines (such as wearing of masks) (e.g., HCW, family of HCW, etc)?

c) Then, what should have been done?

12. What is the major contribution of the current study in view of what is already known?

13. What are the main hypotheses that were generated/tested in the current study?

14. What are the major implications for policy and practice of the current study?

VI. Study limitations:

1. To what end was such "routine health information" planned to be used in the current study?

2. What are the implications of such "routine health information" not being "included in this paper"?

3. How about socio-demographic data of respondents? Can that be provided? Instead of just saying "WRA" or "HCW", could not it be more informative to write "age-WRA" or "age-sex-HCW", etc? If not, is not that a major limitation which should be acknowledged/discussed?

4. How does the limited generalisability affect the utility of the findings?

VII. Conclusions:

1. It is already known that governments need to expand the infrastructure. Perhaps it could be more informative to indicate what should be the priority based on the findings of the current study.

2. What are the lessons learned and how these could be applied to future or other outbreaks or pandemics, such as Ebola disease? For example, see here: https://www.afro.who.int/countries/uganda/news/uganda-defines-priorities-and-needs-its-ebola-response-plan;
https://www.afro.who.int/countries/uganda/news/who-bolsters-ebola-disease-outbreak-response-uganda;
https://www.afro.who.int/countries/uganda/news/uganda-declares-ebola-virus-disease-outbreak.

VIII. Acknowledgments:

1. Why are "Community health workers" who "assisted the research team to mobilize the interviewees." not acknowledged?

6. PLOS authors have the option to publish the peer review history of their article (what does this mean?). If published, this will include your full peer review and any attached files.

**Do you want your identity to be public for this peer review?** For information about this choice, including consent withdrawal, please see our Privacy Policy.

Reviewer #1: No

---

## [Decision Letter · Decision Letter 1]

27 Sep 2023

PGPH-D-22-00887R1

Impact of the COVID-19 pandemic and policy response on access to and utilization of reproductive, maternal, child and adolescent health services in Kenya, Uganda and Zambia.

Dear Dr. Kuria-Ndiritu,

Thank you for submitting your manuscript to PLOS Global Public Health. After careful consideration, we feel that it has merit but does not fully meet PLOS Global Public Health’s publication criteria as it currently stands. Therefore, we invite you to submit a revised version of the manuscript that addresses the points raised during the review process.

We wish to thank the authors for this important piece of work and for the thorough responses provided to address the first round of review.  After careful consideration of the revisions that you made, some minor issues still need  to be addressed. Therefore, we invite you to submit a revised version of the manuscript.

We would appreciate if you could please consider the reviewers’ comments forwarded, and specifically address the following points:

Please clearly justify the research gap addressed by the study in the abstract in addition to the introduction of the paper (reviewer 2)Need for conclusions to be supported by the data:Lines 215-218 generally introduce the theme but then the two data pieces are on family planning and one is unclear what service it is referring to, please add a line specific to family planning to introduce the data or include data to highlight how other services mentioned were also affected (reviewers 1). The representative quotes provide evidence for reduction of clients for family planning services. Are there other quotes or findings to show this impact on all services listed ? (reviewer 2)The author group tried to look at routine HMIS data but there were challenges. Please add this back into the limitations and explain what the limitations to this were (reviewer 1).Reduction in number of clients for a range of services (Line 215 to 216): representative quotes provide evidence for reduction of clients for family planning services. Are there other quotes or findings to show this impact on all services listed ? (reviewer 1 and 2)

We look forward to receiving your revised manuscript.

Kind regards,

Marguerite Massinga Loembe, PhD

Academic Editor

Journal Requirements:

Additional Editor Comments (if provided):

Reviewers' comments:

Reviewer's Responses to Questions

**Comments to the Author**

1. If the authors have adequately addressed your comments raised in a previous round of review and you feel that this manuscript is now acceptable for publication, you may indicate that here to bypass the “Comments to the Author” section, enter your conflict of interest statement in the “Confidential to Editor” section, and submit your "Accept" recommendation.

Reviewer #2: (No Response)

Reviewer #3: (No Response)

2. Does this manuscript meet PLOS Global Public Health’s publication criteria? Is the manuscript technically sound, and do the data support the conclusions? The manuscript must describe methodologically and ethically rigorous research with conclusions that are appropriately drawn based on the data presented.

Reviewer #2: Yes

Reviewer #3: Yes

3. Has the statistical analysis been performed appropriately and rigorously?

Reviewer #2: N/A

Reviewer #3: N/A

4. Have the authors made all data underlying the findings in their manuscript fully available (please refer to the Data Availability Statement at the start of the manuscript PDF file)?

Reviewer #2: Yes

Reviewer #3: Yes

5. Is the manuscript presented in an intelligible fashion and written in standard English?

Reviewer #2: Yes

Reviewer #3: Yes

6. Review Comments to the Author

Reviewer #2: This is an interesting article which contributes to the body of knowledge on the impact of COVID-19 and associated policy responses on access and utilisation of RMNCAH services. The authors also made it clear in the introduction, the study’s main contribution was its focus on different geographical areas that have yet to be explored and comparism of three countries in exploring the perspectives of women of reproductive age, front line health workers and government representatives. Such a clear statement of the research gap this study fills is important, as evidence of the impact of COVID-19 on RMNCAH is already well documented. Perhaps the authors will consider clearly stating this gap in the abstract as well to strengthen it.

Other comments include:

Line 89-90: “Studies have shown mixed results on the impact of the pandemic on RMCAH, especially in sub- Saharan Africa, there is evidence that the pandemic disproportionately affected women and girls in many countries”. Can the authors clarify what they mean by mixed results. Does this mean there are positive impacts or unclear findings on the impact of the pandemic on RMNCAH?

Line 211: “ the quality of care and the lived experiences of the women as they sort services”. Do authors mean sort or sought?

Line 213: The sub-heading is titled effects of the COVID-19 pandemic and the resulting policy responses on access to and utilization of RMCAH services. However, the findings do not show indicate which policy responses in question and the influence on access and utilisation is unclear. Similarly, the findings indicate a reduction in number of clients for a range of services, however the representative quotes provide evidence for reduction of clients for family planning services. Are there other quotes or findings to show this impact on all services listed (Line 215 to 216).

Line 375: Reference made to adolescents in the finding, but the study has excluded this age group and the representative quotes do not reference adolescents. Can this be revised or clarified?

Line 466: “In Uganda, at the onset of the pandemic, the facilities were not offering the youth friendly services as they were considered non-emergency”. Unclear whether it was respondents who considered such services as non-emergency or the providers. Kindly clarify.

Line 597-598: “Most of our findings corroborate the findings of other studies that have now been published”. Can the authors cite these other studies.

Line 600: Kotler is referred to as a single author when the reference includes several authors. It should be rephrased.

Line 609-615: The authors cite studies that have specifically investigated the links between interrupted services and increase in maternal mortality. However, no specific date on the impact of service interruptions on maternal mortality was collected in this study, as such making such inferences seem out of place. Consider rephrasing this section.

Line 616 “Family planning services and Youth friendly services”. Authors have referred to populations in the study as vulnerable, yet it is unclear what makes them vulnerable. Is this based on respondents being young women ( even though the study has specifically excluded adolescent populations)?

Line 637-638: “Similar to our findings on interruption of provision of youth friendly services, Compact for Young People in Humanitarian Action and other studie”. Sentence is incomplete and unclear. Please rephrase or rewrite.

Line 643-644: Can authors enumerate what the “very similar challenges” found in the study as compared to the literature are.

Line 679-683: Unclear how limited information among health workers serves as a policy and system level barrier to RMNCAH services. Similarly, authors highlight efforts to inform community members was hampered but how does this relate to access and utilisation of RMNCAH services.

Overall in the discussion, the authors do a good job in comparing the findings to literature, but the overall meaning of their findings does not come through. They may want to consider this in revisions to make the discussion section stronger.

Reviewer #3: Dear authors, Thank you for undertaking this important piece of work. Overall, you have addressed the comments from the earlier reviewers. I have a few minor additional revisions that I would recommend.

1) Introduction: The introduction feels a bit repetitive and could be more concise, for examples lines 77-79 could be incorporated into the paragraph above, they seem to be one sentence floating alone. Lines 92-97, could come earlier in the introduction - after the opening paragraph when the SDGs are first mentioned.

2) Methods: Table 1 included as supplementary material should be included in the body of the paper, by doing this you will also be able to reduce the text in the methods section by referring to the table. Line 190 - research objectives were referred to but they have not been stated earlier in the paper.

3) Results: Lines 215-218 generally introduce the theme but then the two data pieces are on family planning and one is unclear what service it is referring to, please add a line specific to family planning to introduce the data as is done with ANC lat on or include data to highlight how other services mentioned were also affected. I saw in the limitations section in the tracked changed version that the author group had tried to look at routine HMIS data but there were challenges. Please add this back into the limitations and explain what the limitations to this were. The claim on a reduction in the number of clients attending services would be stronger if it was accompanied by quantitative data. Line 215 'observed' might be better than 'reported.' Line 224 - please include what service is this in reference to. Line 580, a figure with the SEM levels within the context of the study would be helpful to give a summary overview of the findings.

4) General: Please closely review the paper for language and grammar, there are several small typos throughout the paper. This may just be personal preference but 'lived' is not needed to qualify experience when it's clear that it's the experiences and perspectives of the women being interviewed.

7. PLOS authors have the option to publish the peer review history of their article (what does this mean?). If published, this will include your full peer review and any attached files.

**Do you want your identity to be public for this peer review?** For information about this choice, including consent withdrawal, please see our Privacy Policy.

Reviewer #2: No

Reviewer #3: No

---

## [Decision Letter · Decision Letter 2]

30 Nov 2023

Impact of the COVID-19 pandemic and policy response on access to and utilization of reproductive, maternal, child and adolescent health services in Kenya, Uganda and Zambia.

PGPH-D-22-00887R2

Dear Dr. Kuria-Ndiritu,

We are pleased to inform you that your manuscript 'Impact of the COVID-19 pandemic and policy response on access to and utilization of reproductive, maternal, child and adolescent health services in Kenya, Uganda and Zambia.' has been provisionally accepted for publication in PLOS Global Public Health.

However, the editor would like to bring the following spelling issues to the attention of the authors:

Inconsistent use of: RMCAH, RMNCAH, MCAH acronyms, please harmonize throughout manuscript

Page 16, line 313, typo - three not tree

Page 31, line 613, type - believe not belief

Best regards,

Marguerite Massinga Loembe, PhD

Academic Editor

Reviewer Comments (if any, and for reference):

Reviewer's Responses to Questions

**Comments to the Author**

1. If the authors have adequately addressed your comments raised in a previous round of review and you feel that this manuscript is now acceptable for publication, you may indicate that here to bypass the “Comments to the Author” section, enter your conflict of interest statement in the “Confidential to Editor” section, and submit your "Accept" recommendation.

Reviewer #2: All comments have been addressed

Reviewer #3: All comments have been addressed

2. Does this manuscript meet PLOS Global Public Health’s publication criteria? Is the manuscript technically sound, and do the data support the conclusions? The manuscript must describe methodologically and ethically rigorous research with conclusions that are appropriately drawn based on the data presented.

Reviewer #2: Yes

Reviewer #3: Yes

3. Has the statistical analysis been performed appropriately and rigorously?

Reviewer #2: N/A

Reviewer #3: N/A

4. Have the authors made all data underlying the findings in their manuscript fully available (please refer to the Data Availability Statement at the start of the manuscript PDF file)?

Reviewer #2: Yes

Reviewer #3: Yes

5. Is the manuscript presented in an intelligible fashion and written in standard English?

Reviewer #2: Yes

Reviewer #3: Yes

6. Review Comments to the Author

Reviewer #2: This revision provides a much clearer articulation of the research gap in the abstract and clarity of the study findings around policy responses. The discussion has also improved to provide meaning and interpretation of study findings.

Reviewer #3: Thank you for addressing the comments. I noticed two small typos. I would suggest reviewing again for any others before finalising.

Page 16, line 313, typo - three not tree

Page 31, line 613, type - believe not belief

7. PLOS authors have the option to publish the peer review history of their article (what does this mean?). If published, this will include your full peer review and any attached files.

**Do you want your identity to be public for this peer review?** For information about this choice, including consent withdrawal, please see our Privacy Policy.

Reviewer #2: No

Reviewer #3: No
